# Efficacy of aerial forward-looking infrared surveys for detecting polar bear maternal dens

Tom S. Smith[1]*, Steven C. Amstrup[2,3], B. J. Kirschhoffer[2], Geoffrey York[2]

1 Wildlife and Wildlands Conservation Program, Brigham Young University, Provo, UT, United States of America, 2 Polar Bears International, Bozeman, MT, United States of America, 3 Department of Zoology and Physiology, University of Wyoming, Laramie, WY, United States of America

☯ These authors contributed equally to this work.
* tom_smith@byu.edu

**Data Availability Statement:** All relevant data are within the paper.

**Funding:** The author(s) received no specific funding for this work.

## Abstract

Denned polar bears (*Ursus* maritimus) are invisible under the snow, therefore winter-time petroleum exploration and development activities in northern Alaska have potential to disturb maternal polar bears and their cubs. Previous research determined forward looking infrared (FLIR) imagery could detect many polar bear maternal dens under the snow, but also identified limitations of FLIR imagery. We evaluated the efficacy of FLIR-surveys conducted by oil-field operators from 2004–2016. Aerial FLIR surveys detected 15 of 33 (45%) and missed 18 (55%) of the dens known to be within surveyed areas. While greater adherence to previously recommended protocols may improve FLIR detection rates, the physical characteristics of polar bear maternal dens, increasing frequencies of weather unsuitable for FLIR detections—caused by global warming, and competing false positives are likely to prevent FLIR surveys from detecting maternal dens reliably enough to afford protections consonant with increasing global threats to polar bear welfare.

## Introduction

Polar bears (*Ursus* maritimus) construct snow dens in which they give birth to and nurture altricial young [1]. In arctic Alaska, dens in drifted snow are excavated from mid-October through early December. Cubs are born in midwinter [1], and family groups abandon dens by mid-April and head to areas where seal hunting can resume [2, 3].

The geographic scope of petroleum exploration and development in the Alaskan Beaufort Sea coastal region has been expanding, and is now proposed for the Coastal Plain of the Arctic National Wildlife Refuge [4] which is designated critical polar bear denning habitat. Simultaneously, the proportion of maternal polar bears choosing to den on land has been increasing for the past 2 decades [1, 5]. Therefore, the likelihood that maternal dens could be disturbed can be expected to increase. Because polar bear cubs cannot leave the shelter of the den until approximately 3 months of age [6], disruption of denning can have negative consequences for cubs and maternal females [2, 7]. Additionally, industry-related den disturbance can have

**Competing interests:** The authors have declared that no competing interests exist.

significant economic consequences, including: rerouting roads, delays in exploration and production, fines, and other penalties.

Maternal dens usually remain unopened through winter and are essentially invisible under the snow. Previous research determined forward looking infrared (FLIR) imagery could detect the temperature differential between snow over some polar bear maternal dens and the snow where dens were absent [8, 9, 10]. That research, however, also identified limitations of FLIR imagery, and recommended "best practices" protocols to maximize detection abilities. To detect and hence avoid disturbances to maternal dens, oil companies operating in southern Beaufort Sea coastal areas of northern Alaska began using FLIR in 2004 to locate polar bear dens within oil-field operating areas so they can be avoided during ice road construction and other exploration and production activities.

The purpose of this study was to evaluate the influence of environmental variables (e.g., snow depth, wind, relative humidity, etc.) on the efficacy of industry-operated aerial FLIR surveys (hereafter referred to as "industry AFS") and to make recommendations for future den detection and avoidance efforts.

## Study area

The study area included northern Alaska's Beaufort Sea coastal areas (commonly called the North Slope), extending 133 km west and 91 km east of Prudhoe Bay (70˚20' N, 148˚24' W, Fig 1). The Prudhoe Bay region has a semi-arid-tundra climate. The mean annual temperature at Deadhorse, an unincorporated community providing airport services, weather observations and an unofficial hub for Prudhoe Bay operations, is −11˚C (12˚F). The warmest month, July, has a daily average temperature of 8.3˚C (47˚F), the coldest month February at -28˚C (-18˚F; [11]).

Alaska's North Slope lacks the steep topography associated with other denning areas such as Wrangel and Herald Islands, Russia [12, 13], and Svalbard, Norway [14]. The predominantly flat topography of coastal arctic Alaska means suitable denning habitat is restricted to riverbanks, coastal bluffs, barrier islands and other areas where relief is sufficient to catch drifting snow [1, 2, 15, 16, 17]. Amstrup [6] reported > 80% of dens, located by radio telemetry along Alaska's north slope, were within 10km of shore, but a small number have been located as far inland as 50 km [16].

## Methods

We assessed the efficacy of industry AFS for polar bear den site detection by comparing AFS data from 2004–2016 with ground-truth data we collected during research we conducted on emergence behaviors of denning polar bears between 2002 and 2016 in the same area [3, 18]. We monitored maternal dens by direct observation and by placement of video cameras that recorded bear activity. In this manner we were able to ascertain the precise timing of den opening, duration of stay post-breakout, and when family groups abandoned dens [3, 18].

Industry AFS were conducted with the Star Safire (models II, III, and HD 380) FLIR camera units (www.flir.com). FLIR units employed were gimbal-mounted (single axis rotational support) under a de Havilland DHC-6 Twin Otter for all surveys referenced here. This mounting system allowed the FLIR imager to be directed independent of altitude and in any direction below the horizontal plane of the aircraft. The Safire, operates in the 8 to 14 micron wavelength range, and under ideal circumstances can detect temperature differences as small as 0.1˚C [10]. Here we refer to thermal signatures detected by industry AFS as "hot spots." When a hot spot was detected, observers changed FLIR camera view angle, aircraft altitude and position in an attempt to determine whether it was a den or some other source of heat. Numerous

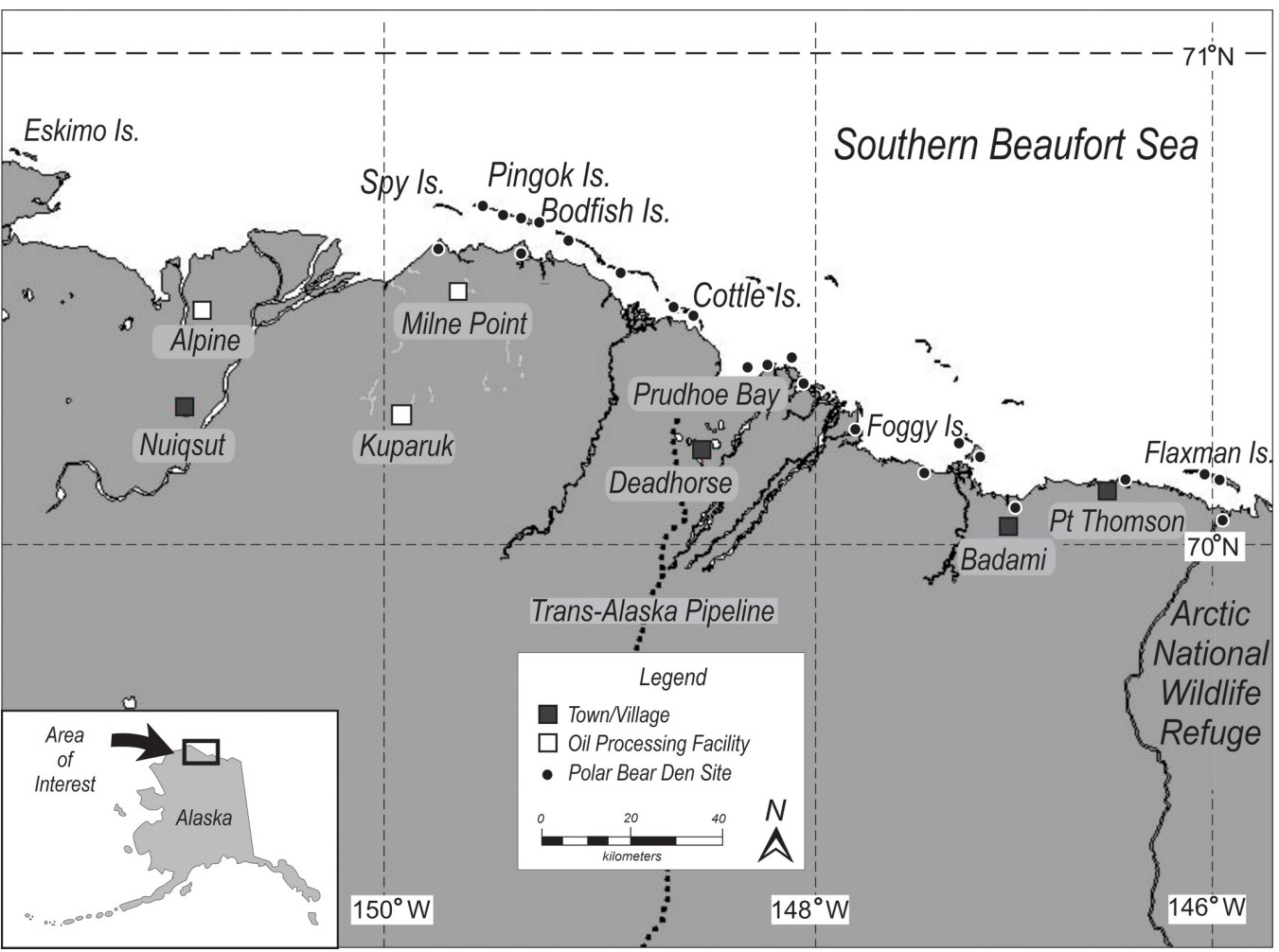

**Fig 1. Study area where FLIR aerial surveys and ground truthing of polar bear den detection surveys were conducted in northern Alaska, 2004–2016.**

potential targets in this environment (e.g. cracks in sea ice, exposed soil, large rocks, or some manmade objects like abandoned 55-gallon steel drums) collect and reradiate heat differently than snow covered ground, and can emit thermal signals similar to those from dens (Fig 2). Hotspots with the "right" signature [9], detected during surveys, were marked as putative maternal dens. In general, den-associated hotspots were several meters in diameter and warmer than the surrounding terrain by at least 10 ºC (T. Smith personal observation). FLIR generated video, including observer audio comments about detected hotspots, was recorded for each flight and archived. Surveyors recorded weather conditions, reported by the flight service station in Deadhorse, during each survey. To supplement weather data recorded during industry AFS, we collected data from the closest ground-based weather stations to each survey flight for the years 2004–2007. These data included ambient temperature, wind speed, relative humidity, and dew point. Dew point incorporates the effect of pressure and temperature on relative humidity.

Personnel involved in industry AFS provided us with summarized survey reports for the years 2008–2016. Those reports provided dates, times, prevailing weather conditions, number of putative dens recorded, and areas in which no dens were observed. Prior to 2008, we received only putative den locations and observations of other polar bear signs.

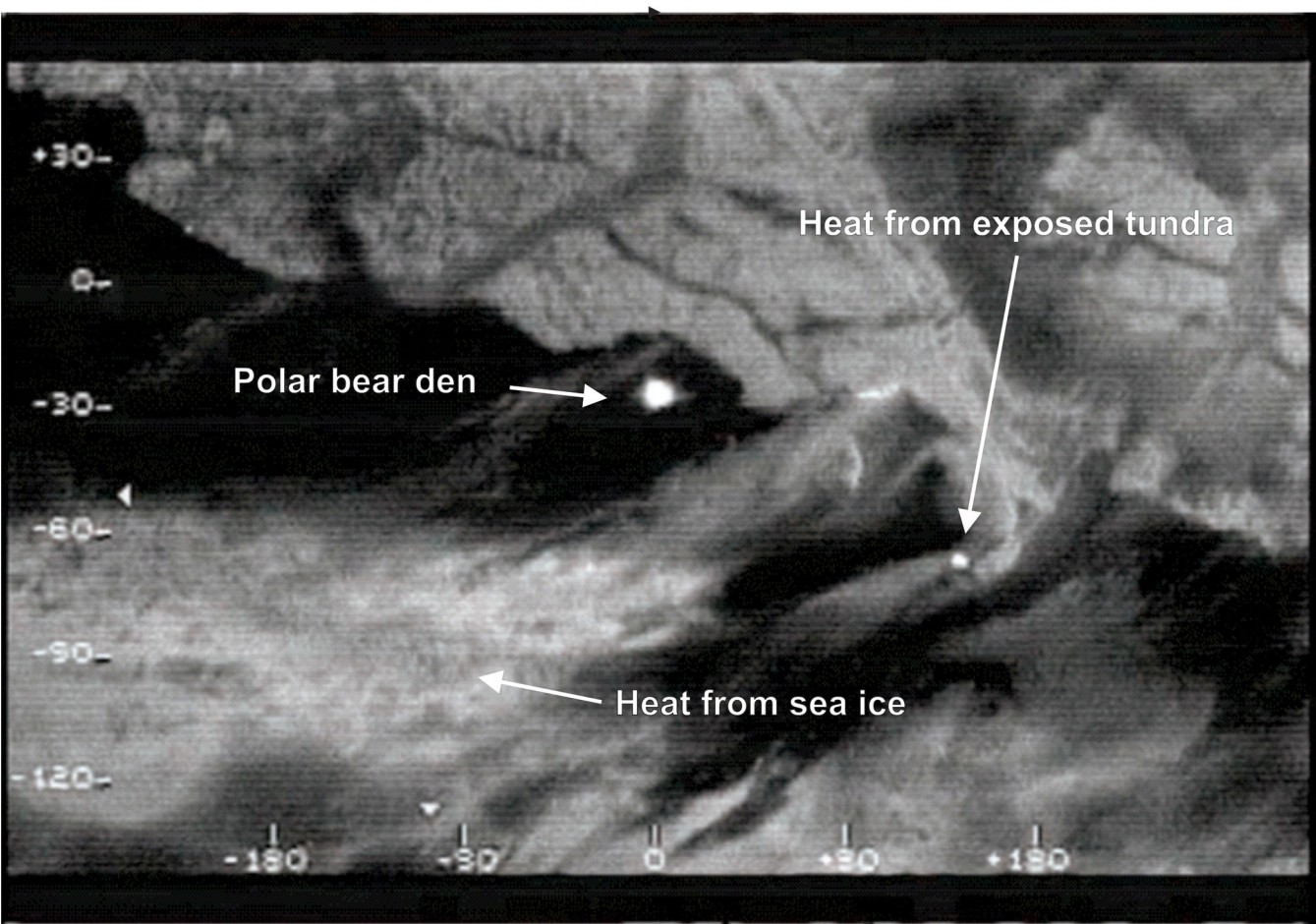

**Fig 2. Forward-looking infrared image of two polar bear dens in the snow bank on the south shore of an Alaskan coastal island.** Also note hotspots created by exposed tundra and warmth radiating up from sea ice with a thin covering of snow.

Along with putative den locations from industry AFS, we used a hand-held FLIR imager (ThermaCAM P65 HS, FLIR Systems) with a 72mm infrared telephoto lens to identify potential den sites by visiting historically high-use denning areas by snow machine and scanning snowbanks with the hand-held unit. Additionally, den locations of radio-collared polar bears were provided by the U.S. Geological Survey (USGS) and by the US Fish and Wildlife Service (FWS) that had been confirmed by visual observation or the use of Karelian Bear Dogs. We monitored putative den locations provided by industry AFS, to ascertain which ones were correctly identified as maternal dens. In addition to positive FLIR identifications of den sites, we tabulated the number of dens known to be within surveyed areas that were missed by industry AFS (false negatives), and we evaluated the frequency of hotspots that were incorrectly identified as dens but proved otherwise (false positives). After bears left their dens in spring, we visited each den site and recorded the depth of snow overlying the main chamber, and other site characteristics consistent with work conducted by Durner [16]. This research was conducted under Federal Fish and Wildlife Permit #MA225854-1, issued by the US Fish and Wildlife Service, Division of Management Authority, Washington, D.C.

## Results

Once den breakout began in the study area (mean date = 16 March, range = 1 March to 4 April; Fig 3), den locations could be verified visually because mounds of excavated snow and tracks and other signs of bear activity surrounding den openings were visible on the snow surface. Consequently, we believe our thorough ground assessment of the survey area provided an accurate evaluation of industry AFS results.

Between 2004 and 2016, we identified 33 maternal polar bear dens within areas also surveyed by industry. Of those 33 dens, 15 (45%) were detected by industry AFS (Table 1). Industry AFS operators also identified 19 putative polar bear dens that our field work proved to be thermal signatures generated by other sources. Because aerial inspection of these "false positives" during industry AFS, proved insufficient to differentiate them from actual dens, monitoring and activity restrictions required for maternal dens also were required for these locations. Prevailing weather at the time of each AFS is presented in Table 2. Weather variable values highlighted in **bold** in Table 2 indicate conditions that diminish and/or preclude den detectability by FLIR [8, 9, 10]. No industry AFS den surveys were conducted when the sun was above the horizon.

## Discussion

During 13 years of industry AFS only 15 of 33 (45% detection rate) polar bear dens known to be within the areas surveyed were detected. We considered late denning as a possible

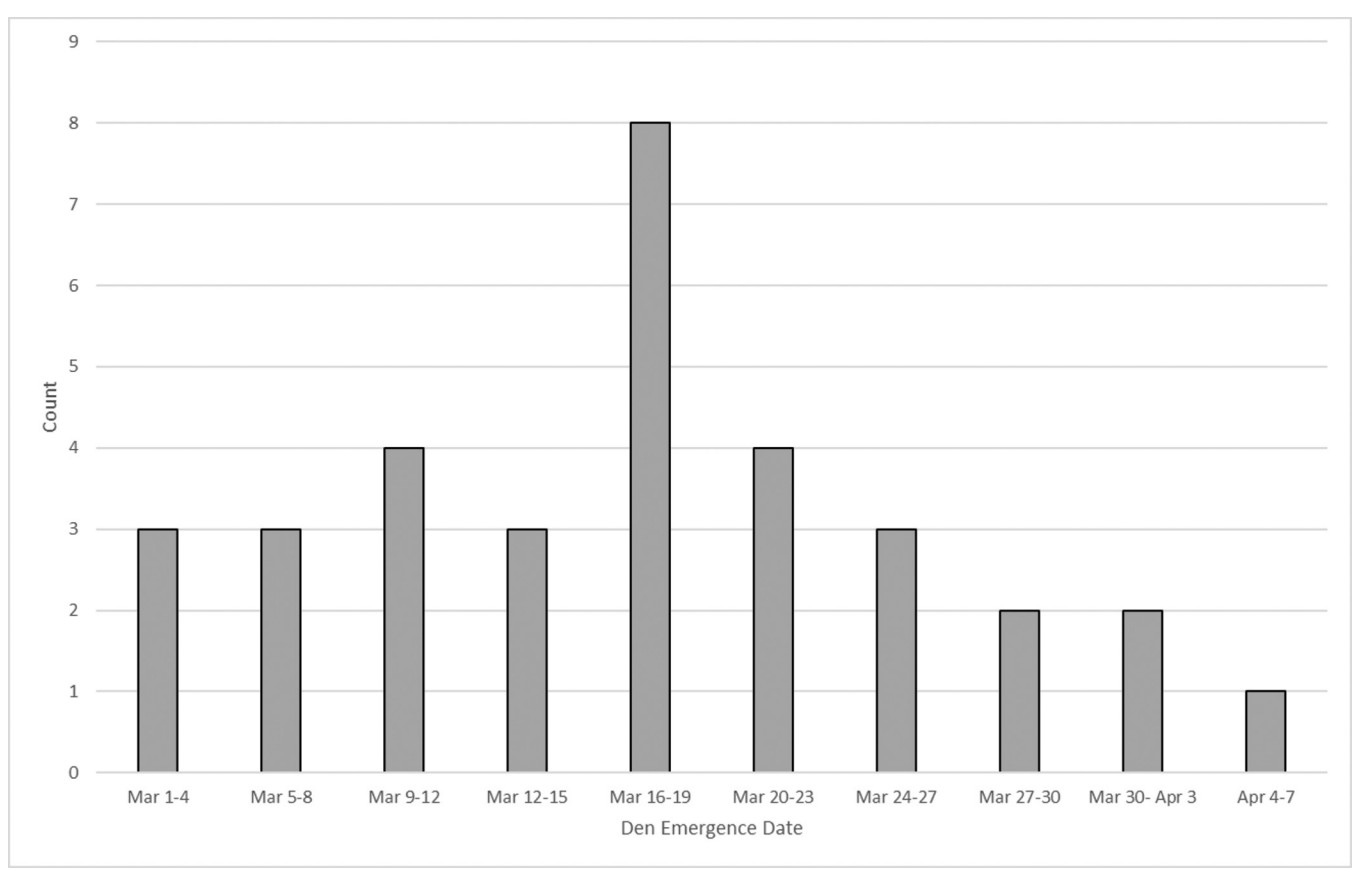

**Fig 3. Emergence dates for 32 polar bear dens within the study area from 2002–2014.** Emergence time for one den discovered during this study was not accurately known.

**Table 1. Summary of observations collected during aerial FLIR surveys to detect polar bear dens in northern Alaska, 2004 to 2016.**

| Survey Year | False Positives[1] | False Negatives[2] | Positives[3] | Total Dens in Survey Area[4] |
|---|---|---|---|---|
| 2004 | 3 | 1 | 0 | 1 |
| 2005 | 0 | 6 | 1 | 7 |
| 2007 | 3 | 1 | 3 | 4 |
| 2008 | 2 | 4 | 2 | 6 |
| 2009 | 3 | 2 | 4 | 6 |
| 2010 | 0 | 1 | 0 | 1 |
| 2011 | 2 | 0 | 2 | 2 |
| 2012 | 4 | 0 | 0 | 0 |
| 2013 | 0 | 1 | 0 | 1 |
| 2014 | 1 | 1 | 0 | 1 |
| 2015 | 0 | 0 | 2 | 2 |
| 2016 | 1 | 1 | 1 | 2 |
| Total dens (% total) In AFS surveyed area | | 18 (55%) | 15 (45%) | 33 |
| Total AFS Observations | 19 (36%) | 18 (35%) | 15 (29%) | 52 |

[1] All hotspots that were thought to be dens but proved otherwise.

[2] Areas cleared by AFS as not having bear dens present but researcher surveys later proved that they had them.

[3] All hotspots that were thought to be dens and were proven correct.

[4] As determined by researchers on snow machines who surveyed areas frequently ($< 4$ d) and used cameras to document presence/absence of bears.

explanation for the failure of industry AFS to detect dens. However, Amstrup and Gardner [1] reported November 11 as the mean den entrance date for polar bears denning successfully on land. Rode et al. [19] also reported a November 11 mean den entrance date (SD of 18.5 d) for polar bears in northern Alaska. While it is possible for bears to have entered dens after survey dates presented in Table 2, nearly 95% of all bears were denned before 4 December (USGS 2018)—the earliest recorded industry AFS effort. Hence ambient conditions and other limitations of FLIR are more likely drivers of the low detection rate of industry AFS.

**Table 2. Prevailing weather conditions during FLIR aerial survey flights for polar bear den detection, northern Alaska, 2007–2016.** We were not provided weather data for surveys conducted prior to 2007.

| Survey Date | Temperature (°C) | Dew point/temp spread (°C) | Wind Speed (km/h) | Misidentified Den Sites |
|---|---|---|---|---|
| 4 Dec 2007 | -15 | 5 | 9.7 | 3 |
| 29 Jan 2008 | -24 | 19 | **29.0**[1] | 3 |
| 8 Dec 2008 | -22 | no data | **33.8** | 3 |
| 7 Dec 2009 | -8 | 12 | 0.2 | 5 |
| 17 Dec 2011 | -24 | 3 | **32.2** | 2 |
| 6 Jan 2012 | -27 | 3 | **16.1** | 4 |
| 14 Dec 2012 | -38 | no data | 9.7 | 1 |
| 9 Dec 2013 | -15 | **2** | **48.3** | 3 |
| 3 Dec 2014 | -11 | 7 | 3.2 | 1 |
| 4 Dec 2014 | -6 | 7 | 6.4 | 1 |
| 5 Dec 2014 | 3 | 5 | **16.1** | 1 |
| 6 Dec 2014 | -8 | 5 | **25.7** | 2 |
| 4 Dec 2015 | -28 | 3 | **16.1** | 2 |
| | = -17 | = 6.5 | = 19.0 | Total = 31 |

[1]Values in bold are weather conditions precluding FLIR detection.

In Table 2, weather variables that are outside the optimal operational window for FLIR are highlighted in **bold.** Eight of thirteen flights were conducted with winds (> 10 km/h) that likely obscured thermal signatures of dens. During these flights, twenty polar bear dens were misidentified (false positives and/or false negatives). Two other flights, close to the wind speeds that block den detection, had four misidentified den sites (Table 2). One flight was conducted below the recommended temperature-dew point spread of 2.8 ºC, as recommended by Amstrup et al. [8]. Flying surveys outside of the recommended weather windows prescribed by York et al. [8], Amstrup et al. [8], and Robinson et al. [10] diminishes chances of den detection. That, in conjunction with snow covering dens possibly being > 1 m (Robinson et al. [10]), may account for the detection rates reported in Table 1.

To test the ability of FLIR to detect dens, Amstrup et al. [8] flew multiple surveys over 23 denning bears for which exact locations were known by radio-telemetry. Only 7 of the 23 dens (30%) were detected on every flight, and 4 dens (17%) were not detected, underscoring the importance of repetitive surveys. However, between the years 2004–2016, AFS were conducted without repetition as recommended by Amstrup et al. [8]. Weather conditions (e.g., wind, precipitation, temperature-dew point spread) and conducting surveys in daylight, or too soon after snow fall or drifting caused by high wind, were identified as principal reasons for detection failure. Robinson et al. [10] advised against conducting surveys during daylight hours and Amstrup et al. [8] concluded the probability of detecting dens in sunlight was essentially zero. None of these AFS surveys were conducted with the sun above the horizon. Robinson et al. [10] also reported that when wind was > 10 km/h den detection with FLIR was increasingly unlikely. Of the industry AFS listed in Table 2, 42% (5 of 12) were conducted with winds < 10 km/h. However, of those five surveys, two were conducted with prevailing winds very close to the detectability cutoff (9.7 km/h). Hence, only 23% (*n* = 3) of industry AFS, for which we have data on ambient conditions, were performed under wind conditions that were conducive to den detection with FLIR. However, those surveys included seven misidentified den sites (Table 2), which could have been due to factors other than wind (e.g., excessive snow depth over dens, objects generating false positives such as rocks, discarded barrels, exposed tundra).

Although ambient conditions at the time of surveys appear to explain much of the detection failure rate; the 4 dens Amstrup et al. [8] did not detect were visited 6 times, including FLIR surveys conducted under optimal weather conditions. Variable snow depth over maternal dens may explain detection failures occurring even when survey conditions seem appropriate. Robinson et al. [10] reported that even under ideal ambient conditions, hand-held FLIR was unable to detect a thermal signature emanating from artificial test dens if roof thickness exceeded 90 cm. While the actual roof thickness that precludes detection by industry AFS is uncertain, it is clear there is a snow depth threshold that prevents sufficient heat generated by denned bears from reaching the snow surface. Durner et al. [16] reported the mean den roof thicknesses for 22 polar bear dens in northern Alaska was 72 ± 87 cm, and ranged from as little as 10 cm to more than 400 cm. Snow depth over many dens, therefore, is likely near, or above, the limits of FLIR detection capabilities, regardless of weather—corroborating the conclusion of Amstrup et al. [8] that some dens will not be detected with FLIR. Finally, Amstrup et al. [8] recommended helicopters be used over fixed-wing aircraft since they can hover, thus allowing a better look at thermal features from a variety of angles and for an extended period of time.

Mid-winter den abandonment may account for some putative den locations that we found not to be actual dens. More hotspots not associated with den emergence (36% of all putative den sites, Table 1), were reported by industry AFS than the actual number of dens detected. This far exceeds documented mid-winter den abandonments (mean = 12% across all years, [20]). York et al. [9] advised that hotspots of interest should be revisited on subsequent days under good environmental conditions to confirm whether they are truly dens. The rate of

these "false positive" signals far exceeds anything that could be expected because of mid-winter den abandonment, suggesting industry AFS did not adhere to protocols known to minimize false positives.

## Conclusions

The U.S. Fish and Wildlife Service Conservation Management Plan [21] recognizes the need for "on the ground" protections to assure as many polar bears as possible persist until sea ice is stabilized. The steep decline (~40% between 2000 and 2010) in the Southern Beaufort Sea polar bear population was driven by reduced survival, particularly of cubs [22]. This makes clear that maximizing cub survival potential is essential for polar bears in this region to persist. A critical step toward maximizing cub survival is protecting maternal dens. In regions where denning activity overlaps with intensive human activities like oil and gas development, protection of maternal dens begins with knowing where they are. By that measure, industry AFS has fallen short of the protections the threatened southern Beaufort Sea polar bear population requires.

The poor den detection rate of industry AFS is likely due to a combination of factors, including weather-related variables (e.g., wind, temperature-dew point spread, precipitation; Table 2), time of day, and den roof thickness. The latest generation of FLIR imagers, while more advanced and sensitive than those used in the earlier surveys, will still struggle with the fundamental physics of detecting subsurface heating associated with dens in the presence of strong winds, direct solar radiation, falling or blowing snow, or too deep snow overlaying the den (R. Overstreet, FLIR Sales Engineer, personal communications, 21 February 2019). Given current limitations, FLIR is unlikely to assure that all maternal dens can be located and hence protected. In addition, as the Arctic has warmed airborne moisture has increased and is expected to continue to do so [23]. Because airborne moisture significantly limits FLIR detection capabilities, finding optimal operational windows for FLIR surveys is likely to become increasingly difficult, and probabilities of den detection by FLIR may decline.

To maximize detection rates in the near term, industry AFS should consistently follow recommended survey protocols developed during previous research [8, 9, 10]. Also, to the extent possible, industry should continue to conduct AFS in early winter (i.e., early December), when snow accumulation over dens is likely to be at minimums. Synthetic aperture radar (SAR) has shown promise for detecting dens, and is not vulnerable to the weather and daylight constraints that limit FLIR application. Significant testing, however, is required to properly evaluate the promise of SAR. Regardless of whether SAR surveys are a viable solution, the current and future threats to polar bears mandate development of new den location technologies with higher detection rates than possible with FLIR.

## Acknowledgments

We would like to thank anonymous reviewers for their thoughtful comments regarding this manuscript. We would also like to thank the following persons for assisting data collection: S. Partridge, T. D. DeBruyn, J. Wilder, T. Evans, S. Schliebe, C. Perham, R. Shideler, B. Jessop, J. Olson, R. Robinson, W. G. Larson and J. Whiting. Additionally, we thank W. J. Streever, D. Sanzone, and C. Pohl of British Petroleum Exploration (BPX) for their support and assistance, without which this work would not have been possible. Also important was R. Murray and N. Hermon of Alaska Clean Seas, and S. Gogosha and D. Heebner of HilCorp Energy Company who provided invaluable assistance at the Milne Point facility. We extend special thanks to D. Herron and J. Thompson of the BP Thermographics Division for providing access to the FLIR cameras. Polar Bears International (PBI) has played a critical role in supporting this work.

## Author Contributions

**Conceptualization:** Tom S. Smith, B. J. Kirschhoffer, Geoffrey York.

**Data curation:** Tom S. Smith.

**Formal analysis:** Tom S. Smith, Steven C. Amstrup, Geoffrey York.

**Investigation:** Tom S. Smith, Steven C. Amstrup, B. J. Kirschhoffer, Geoffrey York.

**Methodology:** Tom S. Smith, Steven C. Amstrup, Geoffrey York.

**Project administration:** Tom S. Smith.

**Resources:** Geoffrey York.

**Supervision:** Steven C. Amstrup.

**Writing – original draft:** Tom S. Smith, Geoffrey York.

**Writing – review & editing:** Tom S. Smith, Steven C. Amstrup, B. J. Kirschhoffer, Geoffrey York.

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
