## [Decision Letter · Decision Letter 0]

8 Oct 2019

PONE-D-19-24951

Efficacy of aerial forward-looking infrared surveys for detecting polar bear maternal dens

PLOS ONE

Dear Dr. Smith,

Thank you for submitting your manuscript to PLOS ONE. After careful consideration, we feel that it has merit but does not fully meet PLOS ONE’s publication criteria as it currently stands. Therefore, we invite you to submit a revised version of the manuscript that addresses the points raised during the review process.

I believe that we secured 2 excellent reviews and you should have no difficulty making the revisions recommended by the reviewers.  Please provide a detailed cover letter indicating exactly which revisions were made in response to the reviewer comments.  The manuscript is not formally accepted yet, but if you do a conscientious job addressing these comments I will be happy to accept the paper.

We would appreciate receiving your revised manuscript by Nov 22 2019 11:59PM. To enhance the reproducibility of your results, we recommend that if applicable you deposit your laboratory protocols in protocols.io, where a protocol can be assigned its own identifier (DOI) such that it can be cited independently in the future. For instructions see: http://journals.plos.org/plosone/s/submission-guidelines#loc-laboratory-protocols

We look forward to receiving your revised manuscript.

Kind regards,

Mark S. Boyce, Ph.D.

Academic Editor

PLOS ONE

Journal Requirements:

1. In your Methods section, please provide additional information regarding the permits you obtained for the work. Please ensure you have included the full name of the authority that approved the field site access and, if no permits were required, a brief statement explaining why.

Reviewers' comments:

Reviewer's Responses to Questions

**Comments to the Author**

1. Is the manuscript technically sound, and do the data support the conclusions?

Reviewer #1: Partly

Reviewer #2: Yes

2. Has the statistical analysis been performed appropriately and rigorously? 

Reviewer #1: No

Reviewer #2: Yes

3. Have the authors made all data underlying the findings in their manuscript fully available?

Reviewer #1: Yes

Reviewer #2: No

4. Is the manuscript presented in an intelligible fashion and written in standard English?

Reviewer #1: Yes

Reviewer #2: Yes

5. Review Comments to the Author

Reviewer #1: Overall, I think this research is valuable and draws important conclusions, but there is a disjoint throughout that hampers the clarity of the main message. There are details of some sections that seems to be lacking, or details that seem to be added unnecessarily. More structure, continuity across sections, and attention to those details could make the interpretation of the research much clearer. I also suggest some more statistical analysis of the collected data to support the conclusions (described in more detail below).

Suggested Edits

Line 40-42 – The in-text citation should not be a URL. That should be in the Reference section.

Line 44 – Move “therefore” to the beginning of the sentence to increase flow.

Line 59-61 – You should add that you looked at weather and snow depth in your objective (or even just environmental variables), to set up the specific direction of the study early on.

Line 72 – Figure 1- different symbology is used for some municipalities, so I suggest a legend denoting what each means.

Line 88-89 – This sentence is redundant as it is restating what was said in the prior sentence.

Line 91-92 – You are either missing a closing bracket, or you have an extra opening bracket before the URL.

Line 103 – Is there a way you can explain what the “right” signature looks like? Or cite a description from another paper. The lack of description would make reproducibility difficult.

Line 105-106 – This would be a good place to mention which years you were not given weather data for.

Line 114 – Personnel instead of personal.

Methods – You talk about collecting weather data and snow depth data, but you don’t appear to do any further analyses with them. It would be interesting to include an analysis on these data and present any relationships between weather and den detection, or snow depth and den detection in the results section.

Line 132 – You mention monitoring den locations to confirm maternal dens in the methods, but you don’t directly speak to recoding the timing of den break-out. If you present the results, you should mention the methodology to keep all aspects of your study clear to the reader.

Results – Presentation of weather via table is good but does not tell the reader very much about the effects weather may have had on accurate den detection. If there are any prevailing relationships between weather variables and misidentified den sites it would be useful to summarize them in the text.

You collected snow depth, so I would also love to see some numbers on how snow depth affected industry AFS den detection.

If you do this, I would also suggest adding a sentence about these results into your abstract.

Discussion – You frequently mention the results from other work on the effects of snow and weather on den detection, but you don’t speak to it with your own data. You have the data, so you should explore these relationships and present them alongside the previous work you are citing (ie. Robinson et al. 2014 and Amstrup et al. 2004). You told us the results of industry AFS are not sufficient for detection, but how do they compare to previous work in terms of weather, snow, etc.?

Line 154-163 – You present a possible explanation, then immediately strike it down. The information here is valuable, but could you frame it in a way that is just providing a reason to discount the idea that bears entered dens late, without presenting it as a possibility first.

Line 164 – 169 – This would be a good place to compare or contrast the industry results to the results from Amstrup et al. 2004. Return focus to your results, rather than on the results from previous work.

Line 171 – Mention that industry AFS followed this recommendation and only did surveys when the sun was down.

Line 177 – Once you say 3 surveys were done under the correct wind conditions, you should briefly summarize if those 3 had accurate detection.

Line 179 – Change “never detected” to “did not detect”.

Line 182 -190 – Again, in the methods you say you collected snow depth data, it would be very beneficial to include that in your analysis and results, and then discuss your results alongside the other publications you are discussing here. You mention “roof thickness that precludes detection by industry AFS is uncertain” but you have not included this in any results.

Line 195 – Too many spaces after the period, make sure spacing is the same throughout.

Line 197 – Remove extra period after “dens”.

Line 207 – Change “essential to maximizing” to “essential for maximizing”.

Line 208-210 – Add “In regions” before “where” at the start of the sentence. i.e. In regions where…..

Line 211 – Change to “… southern Beaufort Sea polar bear population requires.”

Line 215 – The location of the (Table 2) reference is misleading, table 2 only contains weather variables, placing it at the end of that list makes it seem like it should contain more.

Line 224 – Remove comma between “surveys” and “is”.

Line 232 – This is the first time you are mentioning helicopters are preferable to fixed wings. This should be included earlier and explained to some extent why helicopters are preferable or cited here.

Line 241 – “the answer” feels too definitive. Change to “a viable solution”

Line 248 – “J. Wilder” is listed twice.

Reviewer #2: This is a fairly straight-forward study that addresses a wildlife management technique of importance in parts of the range of polar bears. Forward looking infrared imagery is a technique used to detect polar bear maternity dens and this study provides a quantitative assessment of the method. The study is well-designed, the data and analyses are appropriate, and it is well written. I have few comments but a few that may improve the final version. Given recent interest in opening more the of the North Slope of Alaska to hydrocarbon exploration/extraction, this is a timely contribution to the literature. The conclusion section could be shortened by a good bit without loss of information.

Detailed issues

20 – I like to see binomial names at first mention - same issue on line 34

26-7 – “previously recommended” – this can be deleted as they aren’t articulated in the abstract so “protocols” is sufficient

29 – “hot spots” has an ecological / biological connotation (e.g., hot spots of biodiversity) so I would reword. I am aware of one denning paper that uses hot spot analyses (spatial ecological method) to identify den areas. Maybe “false positive” or something like that?

34 – a reference on denning ecology would be useful here.

36 – some background for non-polar bear experts – “abandon dens” – where do they go?

39 – use of “even” could be seen as a bias by some. It can be dropped without loss of content.

43 – the temporal context of “increasing” would help put the study in context for non-experts

45 – the “altricial” issue is stated in line 34 – once is enough. Maybe drop it from line 34. I don’t know the difference between “very altricial” and “altricial” – consider dropping “very”.

51 – “Previous research” – requires a citation in this sentence not the following one.

86 – when you give a year range, it is by default a “period” so it’s redundant – brevity is always welcome

109-110 – an explanation of dew point is not useful but you could explain why it is being measured (i.e., what effect it may have on the FLIR detection of dens)

140 – I’m not sure what “accurately” applies to here. It suggests that some dens were inaccurately detected. Can you drop this word? I’m thinking issues of accuracy and precision so the use of the word isn’t clear.

Conclusions

Some of the wording could be tempered a bit. Use of words like “imperiled” (line 211) could be replaces with a more neutral term such as “threatened” (the US classification). Overall, this section could be shortened or remove the “Conclusions” header. A conclusion section shouldn’t equal in length the main discussion (at least in my opinion). I find lines 213-220 largely repetitive with the discussion.

220 – “FLIR will never” – that is a overly broad statement – increases in technology are a possible factor. The current methods and technology cannot detect all dens.

225 – “are likely to decline” to “may decline” – Predicting weather events/conditions is a challenge for climate change models.

225-7 – “other technologies” – this sentence doesn’t add significantly to the manuscript. This point is made again on line 237. Once is too much in my opinion.

228-233 – if you could condense this section it would help as the factors listed weren’t directly tested in the study (i.e., they weren’t use as co-variates in a binomial analysis – detected / not detected). Such an analysis is not necessary (I doubt such an analysis would work based on the samples size).

241-2 – this is the 3rd reference to new technology

6. PLOS authors have the option to publish the peer review history of their article (what does this mean?). If published, this will include your full peer review and any attached files.

Reviewer #1: No

Reviewer #2: Yes: Andrew E. Derocher

---

## [Author Response · Author response to Decision Letter 0]

24 Oct 2019

Other than the uploaded 'response to reviewers' document I have nothing else to add.

---

## [Decision Letter · Decision Letter 1]

7 Jan 2020

Efficacy of aerial forward-looking infrared surveys for detecting polar bear maternal dens

PONE-D-19-24951R1

Dear Dr. Smith,

We are pleased to inform you that your manuscript has been judged scientifically suitable for publication and will be formally accepted for publication once it complies with all outstanding technical requirements.

With kind regards,

Ludek Bartos

Academic Editor

PLOS ONE

Additional Editor Comments (optional):

Reviewers' comments:

Reviewer's Responses to Questions

**Comments to the Author**

1. If the authors have adequately addressed your comments raised in a previous round of review and you feel that this manuscript is now acceptable for publication, you may indicate that here to bypass the “Comments to the Author” section, enter your conflict of interest statement in the “Confidential to Editor” section, and submit your "Accept" recommendation.

Reviewer #1: All comments have been addressed

Reviewer #2: All comments have been addressed

2. Is the manuscript technically sound, and do the data support the conclusions?

Reviewer #1: Yes

Reviewer #2: Yes

3. Has the statistical analysis been performed appropriately and rigorously? 

Reviewer #1: Yes

Reviewer #2: Yes

4. Have the authors made all data underlying the findings in their manuscript fully available?

Reviewer #1: Yes

Reviewer #2: Yes

5. Is the manuscript presented in an intelligible fashion and written in standard English?

Reviewer #1: Yes

Reviewer #2: Yes

6. Review Comments to the Author

Reviewer #1: (No Response)

Reviewer #2: I am fully satisfied with the revised version. I believe the paper is an excellent and timely contribution to the literature.

7. PLOS authors have the option to publish the peer review history of their article (what does this mean?). If published, this will include your full peer review and any attached files.

Reviewer #1: No

Reviewer #2: No

---

## [Editor Report · Acceptance letter]

9 Jan 2020

PONE-D-19-24951R1 

Efficacy of aerial forward-looking infrared surveys for detecting polar bear maternal dens 

Dear Dr. Smith:

I am pleased to inform you that your manuscript has been deemed suitable for publication in PLOS ONE. Congratulations! Your manuscript is now with our production department. 

With kind regards,

on behalf of

Dr. Ludek Bartos 

Academic Editor

PLOS ONE